# Deep Learning-Based Radar Composite Reflectivity Factor Estimations from Fengyun-4A Geostationary Satellite Observations

**Fenglin Sun** [1], **Bo Li** [1,*], **Min Min** [2] **and Danyu Qin** [1]

1   Key Laboratory of Radiometric Calibration and Validation for Environmental Satellites (LRCVES/CMA),
    National Satellite Meteorological Center, China Meteorological Administration (NSMC/CMA),
    Beijing 100081, China; sunfl@cma.gov.cn (F.S.); qindy@cma.gov.cn (D.Q.)
2   Guangdong Province Key Laboratory for Climate Change and Natural Disaster Studies,
    School of Atmospheric Sciences, Sun Yat-sen University, Zhuhai 519082, China; minm5@mail.sysu.edu.cn
*   Correspondence: boli@cma.gov.cn; Tel.: +86-010-58995924

**Abstract:** Ground-based weather radar data plays an essential role in monitoring severe convective weather. The detection of such weather systems in time is critical for saving people's lives and property. However, the limited spatial coverage of radars over the ocean and mountainous regions greatly limits their effective application. In this study, we propose a novel framework of a deep learning-based model to retrieve the radar composite reflectivity factor (RCRF) maps from the Fengyun-4A new-generation geostationary satellite data. The suggested framework consists of three main processes, i.e., satellite and radar data preprocessing, the deep learning-based regression model for retrieving the RCRF maps, as well as the testing and validation of the model. In addition, three typical cases are also analyzed and studied, including a cluster of rapidly developing convective cells, a Northeast China cold vortex, and the Super Typhoon Haishen. Compared with the high-quality precipitation rate products from the integrated Multi-satellite Retrievals for Global Precipitation Measurement, it is found that the retrieved RCRF maps are in good agreement with the precipitation pattern. The statistical results show that retrieved RCRF maps have an R-square of 0.88-0.96, a mean absolute error of 0.3-0.6 dBZ, and a root-mean-square error of 1.2-2.4 dBZ.

**Keywords:** deep learning; algorithm for retrieving radar composite reflectivity factors; Fengyun-4A geostationary satellite

## 1. Introduction

Severe weather always shows the natural characteristics of strong intensity, abruptness, wide distribution, rapid development and evolution, along with large destructive power, thus causing widespread concern [1]. A rapid and effective response to severe weather events, such as strong wind, heavy precipitation and flash floods, is essential to reducing the threats to people's lives and property. The key to forecasting these sudden disastrous weather systems is timely observations with a high spatial resolution, which serves as the foundation of both situational awareness and forecast. As is well known, the Doppler weather radar with high spatio-temporal resolution has become one of the most effective tools for analyzing and nowcasting the meso- and micro-scale weather systems. To date, based on the radar echo data, many well-established and robust methods of monitoring, tracking and extrapolating for severe weather have been well-developed and extensively applied [2–5]. Particularly, reflectivity factors, radial velocity and velocity spectrum width derived from the Doppler weather radar echo data can be utilized to analyze the cloud and hydrometeor distribution of severe weather phenomena more accurately and timely [6,7]. Furthermore, the convection initiation determined by the threshold of radar reflectivity factor >35 dBZ can be used to forecast the occurrences of severe weather events in advance [8–10]. In addition to the real-time weather analysis, the

radar reflectivity data are also employed as initial conditions for high-resolution numerical weather prediction (NWP) models to perform cloud analysis [11,12]. In variational data assimilation systems, radar reflectivity can effectively initialize both model hydrometeors and microphysics parameters associated with a microphysical scheme [13,14]. The assimilation of radar reflectivity data into numerical weather prediction models effectively increases the information of moisture and rainwater information in the cloud, further improving the forecasts of location and intensity of convective weather systems [15,16].

However, most of the world still lacks the observational infrastructure of weather radars, especially in ocean and mountainous regions. This is perhaps the most serious disadvantage of relying on radars to monitor and forecast severe weather events in real-time. In contrast, the geostationary (GEO) satellite imaging system can continuously capture images of the Earth from space, and these images are already well applied for cloud cluster tracking. Satellite observations, especially those from the current GEO meteorological satellite with high-frequency observations for tracking convective clouds, can fill the observation gaps of radar and monitor the rapid generation as well as development of severe weather. Recently, with the rapid development of the latest technology, the new- generation GEO meteorological satellites, such as Fengyun-4A, Himawari-8/9, GEO Operational Environmental Satellite-R, have significantly enhanced their spatio-temporal resolution and increased the spectral detecting channels [17–19]. Therefore, the GEO satellites will play a more essential role in monitoring and nowcasting severe convective weather [20–23]. Actually, the new-generation GEO meteorological satellites have already been used in convective initiation nowcasting [24–26], dynamic structure analysis of super typhoons [27], assimilating the infrared radiances to improve convective predictability, precipitation estimation [28,29], and so on [30].

Note that, in recent years, significant progress has been made in deep neural network (DNN) techniques, which is mainly attributed to the increased amount of available data, better training and predicting model architectures, and the ease of implementation on powerful, specialized hardware such as Graphics Processing Units (GPUs) [31,32]. For the application in weather forecasts, the deep neural network techniques have already been used to forecast the precipitation, severe weather and others based on the observation of radar reflectivity data [33,34]. Some new Gate Recurrent Unit (GRU) models (such as theTrajectory GRU and the Generative Adversarial-Convolutional GRU) beyond Long Short-Term Memory were proposed for precipitation nowcasting [35,36]. Recently, a new deep learning model named the MetNet was also proposed and developed to forecast precipitation up to 8 hours in the future at the high spatial resolution of 1 $km^2$ and the temporal resolution of 2 min. It has been revealed that the MetNet performs better than the optical flow-based models [37].

In this investigation, a deep learning-based algorithm is developed to retrieve the radar composite reflectivity factor (RCRF) maps from the observations of the new-generation GEO satellite of Fengyun-4A. Based on the microphysical properties of cloud top, the RCRF maps retrieved from satellite observations can provide the basic detection and diagnosis data for severe weather in areas not covered by radar. Moreover, the RCRF maps provide an excellent supplement to the existing real-time radar data. The fusion of RCRF maps retrieved from the GEO satellite and radar data at different times makes it possible to monitor the rapidly developing convective systems by reducing the time interval. In addition, the RCRF maps retrieved from Fengyun-4A observations can provide a relatively intuitive understanding on the geographical distribution of intensity of strong convective systems and typhoon cyclones where the coverage of radar data is incomplete.

The remainder of this paper proceeds as follows. Section 2 introduces the GEO satellite, the weather radar data and the precipitation rate data from the Global Precipitation Measurement (GPM) mission. Section 3 describes the detailed algorithm for retrieving the RCRF maps. Section 4 provides the validations and discussions about this new algorithm. Finally, Section 5 presents the main conclusions of this study.

## 2. Data

### 2.1. Interest Fields from Fengyun-4A AGRI

Fengyun-4A is China's first new-generation GEO meteorological satellite, which was successfully launched on December 11, 2016. It carries three new, advanced optical instruments, namely the Advanced Geosynchronous Radiation Imager (AGRI), the Geosynchronous Interferometric Infrared Sounder and the Lightning Mapping Imager. The nadir point of Fengyun-4A/AGRI's is located at 104.7°E, with the observation field covering the East Asian, Australia, Indian Ocean, etc. The spatial resolution of the AGRI ranges from 0.5 km (visible channel) to 4 km (infrared channel) at the nadir point. In addition, regular observations can be acquired by two different scanning operations of 15-min full disk and a 5-min Chinese continental domain. Compared with the previous Fengyun-2 GEO satellite, Fengyun-4A has significantly enhanced its capabilities in weather warning and forecasting, due to its increased radiation channels and spatio-temporal resolution [17]. Moreover, the detection channels of Fengyun-4A are more sensitive to the cloud-top properties (such as phase and particle size) that cannot be obtained by using the limited channel selection of the previous Fengyun-2 [38].

Table 1 shows the sensitivity of interest fields to cloud-top properties from Fengyun-4A/AGRI channels, which is used in models for retrieving the radar reflectivity factor. These parameters are chosen because the radar reflectivity factor is closely related to the cloud microphysical properties and hydrometeor distribution [39–41]. The visible channels have a higher spatial resolution (0.5 km), but they are only applicable during the daytime, while the infrared channels can provide full-time observations. For Fengyun-4A/AGRI, the 1.61 μm channel at the near-infrared band shows a high sensitivity to the ice cloud phase and cloud effective radius with a strong-absorbing effect. The visible channel at 0.65 μm is a weakly absorbing channel sensitive to cloud optical thickness and cloud phase, especially for strong convective clouds [42]. The absorption characteristics of cloud particles at 8.6, 10.8, and 12.3 μm are significantly different for various cloud phases. Due to the different absorption effects of liquid water cloud particles, the difference of brightness temperature is smaller between 10.8 and 12.3 μm than that between 8.6 and 10.8 μm, but the situation is opposite for ice cloud particles [43]. To further evaluate the potential application ability of composite radar data, a comparison test between visible and infrared channels for model sensitivity analysis is also proposed in this study.

$$abeldo_M = abeldo \times sec(\theta) \tag{1}$$

where $\theta$ is the solar zenith angle ($\theta < 70°$).

**Table 1.** Fengyun-4A/AGRI interest fields for models retrieving the radar reflectivity factor.

| No | Interest Fields | Physical Basis | Model Index |
|----|-----------------|----------------|-------------|
| 1 | BT 10.8μm | Cloud-top temperature assessment | Model I |
| 2 | BTD 10.8–6.2μm | Cloud-top height relative to tropopause | |
| 3 | BTD 12.3+8.6–2×10.8μm | Cloud-top glaciation/ phase | |
| 4 | Modified albedo 0.65μm | Cloud optical thickness | Model II |
| 5 | Albedo ratio 0.65/1.61μm | Cloud-top glaciation /phase | |

### 2.2. Weather Radar Data

Based on the interpretation of backscattered echoes, the ground-based weather radars detect convective systems, rainfall intensity and speed by receiving the emitted electromagnetic waves. The RCRF maps used in this study are derived from the China New Generation Weather Radar System (data available at http://10.1.64.154/cimissapiweb/apidataclassdefine_list.action accessed on 1 May 2021 [44]. Composite reflectivity is the

maximum base reflectivity value that occurs in a given vertical column in the radar umbrella. The China New Generation Weather Radars scan in several pre-defined volume coverage patterns. In principle, the radars tilt a fixed elevation angle above the horizontal plane and perform a 360° horizontal sweep, then change the elevation angle, and complete another 360° sweep. Composite reflectivity can generate a plane view of the most intense portions of thunderstorms and can also be compared with the base reflectivity to help users to accurately determine the 3-D structure of a thunderstorm. Currently, the temporal resolution of the operational weather radar data is 6 minutes.

*2.3. GPM IMERG Data*

The integrated Multi-satellite Retrievals for the GPM(IMERG) dataset based on a unified U.S. algorithm provides a high-quality precipitation product based on multi-satellite observations. In this study, the IMERG dataset is primarily used to evaluate the performance of retrieved RCRF in the areas not covered by radar. This gridded fusion dataset has a temporal resolution of 30 minutes and a spatial resolution of $0.1° \times 0.1°$ with the maximum precipitation rate of 50 mm·h$^{-1}$, covering the area between the latitudes of 60°S and 60°N. Technically, by the inter-calibration, fusion and interpolation of several satellite microwave precipitation products (such as NOAA Joint Polar Satellite System-Advanced Technology Microwave Sounder, JPSS-ATMS), together with microwave-calibrated infrared satellite estimates (NOAA GOES-E/W), rain gauge analyses, and other potential precipitation estimators, this uniform and gridded precipitation product is estimated at fine spatio-temporal scales for Tropical Rainfall Measuring Mission and GPM eras over the globe [45]. Note that the freely released IMERG V04A version data will be delayed by about 3-4 months. Consequently, this time lag makes it impossible to support near-real-time storm monitoring and nowcasting applications. However, we can still use this high-quality precipitation dataset to validate the RCRF retrieved in this investigation. For more details about IMERG data, please refer to https://disc.gsfc.nasa.gov/datasets/GPM_3IMERGHHL_06/summary?keywords="IMERG%20late" accessed on 1 May 2021 [46,47].

**3. Methodology**

To develop the retrieving method for RCRF maps, we explore a deep learning method that consists of three modules, including data preprocessing, model training and RCRF validating based on three independent datasets, i.e., a training dataset, a validation dataset and a test dataset (Figure 1). The training dataset is used to train the model by optimizing its learnable parameters with the back-propagation algorithm. To evaluate the performance of model, the independent validation dataset is used to assert its ability to generalize well to unseen data, i.e., to make sure that the model is not overfitting.

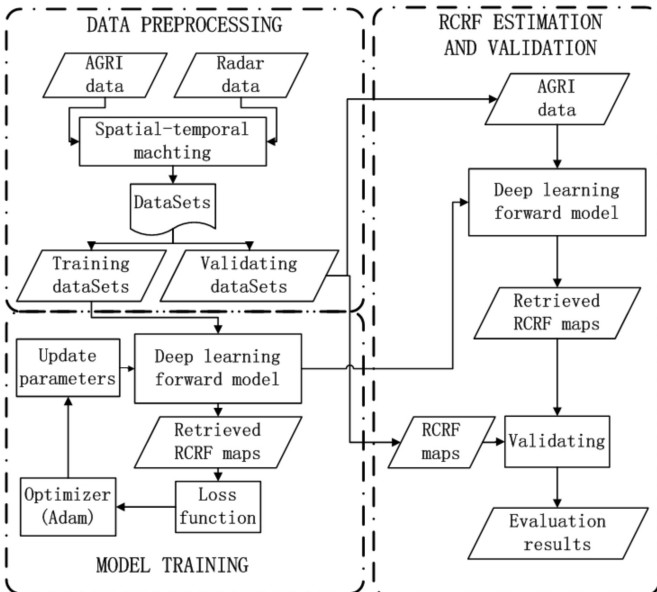

**Figure 1.** Flowchart of the data preprocessing and the training module (**left**) and the RCRF map validating module after training (**right**) of the deep learning model.

*3.1. Training and Validation Data*

To train and estimate RCRF maps by using multi-band infrared/visible observations from Fengyun-4A/AGRI based on a deep learning algorithm, firstly we collected the satellite observations and actual RCRF data for the same period to construct the training and validation datasets. Because of the difference in spatial resolutions between Fengyun-4A/AGRI (about 4.0 km) and the real radar reflectivity factor (about 1.0 km), we averaged the radar reflectivity factor in the horizontal space to match each observed pixel of FY-4A/AGRI. We collocated the simultaneous Fengyun-4A/AGRI and radar reflectivity factor data from May to October in 2020 over a broad spatial coverage in China (70°E–135°E, 16°N–55°N). Since the ground-based weather radars conduct scanning mode within a 6-minute interval, the collocated datasets with an observation time difference should be less than 3 minutes. On the one hand, it ensures that the matching data are one-to-one. On the other hand, it also ensures that the cloud system movement is not obvious within 3 min. The quality control processing for both satellite and radar data is also performed after data matching. After that, we delete the radar echo samples in clear sky pixel determined by Fengyun-4A/AGRI operational cloud mask products [48] and abandon the incomplete satellite data with lost lines. The fully matched dataset randomly selects 70% of samples for model training, 15% for model validation, and the remaining 15% for model testing (Table 2). Since the radar echoes cover a small part of the ocean, this part of data is utilized to validate the samples of the ocean part. The other part of data is treated as the samples of the land part. The complete statistical validation results will be shown in Section 4.2.

**Table 2.** Temporal distribution of the samples for model training and validation.

| Application Month | Model I | | Model II | |
|---|---|---|---|---|
| | Training | Validation | Training | Validation |
| May. | 4284 | 918 | 1927 | 412 |
| Jun. | 4183 | 896 | 1882 | 403 |
| Jul. | 4257 | 912 | 1916 | 410 |
| Aug. | 4273 | 915 | 1923 | 412 |
| Sept. | 4194 | 898 | 1887 | 404 |
| Oct. | 4217 | 903 | 1898 | 406 |

### 3.2. Network Architecture

As illustrated in Figure 2, the network architecture is evolved from the U-net [49]. The U-net has already been widely used in biomedical image segmentation [50,51]. This study introduces a U-net-based regression model that consists of a contracting pyramid structure (left side) and an expansive inverted pyramid one (right side). The contracting structure follows the typical architecture of a convolutional network. It consists of the repeated application of two 3 × 3 convolutions (unpadded convolutions), each followed by a batch-normalization (BN) module, an activation function named rectified linear unit (ReLU) and a 2 × 2 max pooling operation with a stride of 2 for down-sampling. At each down-sampling step, the number of feature channels is doubled. Every step in the expansive architecture consists of an up-sampling of the feature map followed by a 2 × 2 convolution (called an up-convolution) that halves the number of feature channels, a concatenation with the correspondingly cropped feature map from the contracting structure, and two 3 × 3 convolutions, each followed by a BN and a ReLU. At the final layer, a 1 × 1 convolution layer followed by a ReLU is used to map each 16-component feature vector to the desired regression map. Finally, the network has 23 independent convolutional layers and 1,765,440 free parameters in total. Note that the sizes of input data are 800 × 1280 × 3 for the infrared model (model I) and 800 × 1280 × 2 for the visible model (model II) in this investigation.

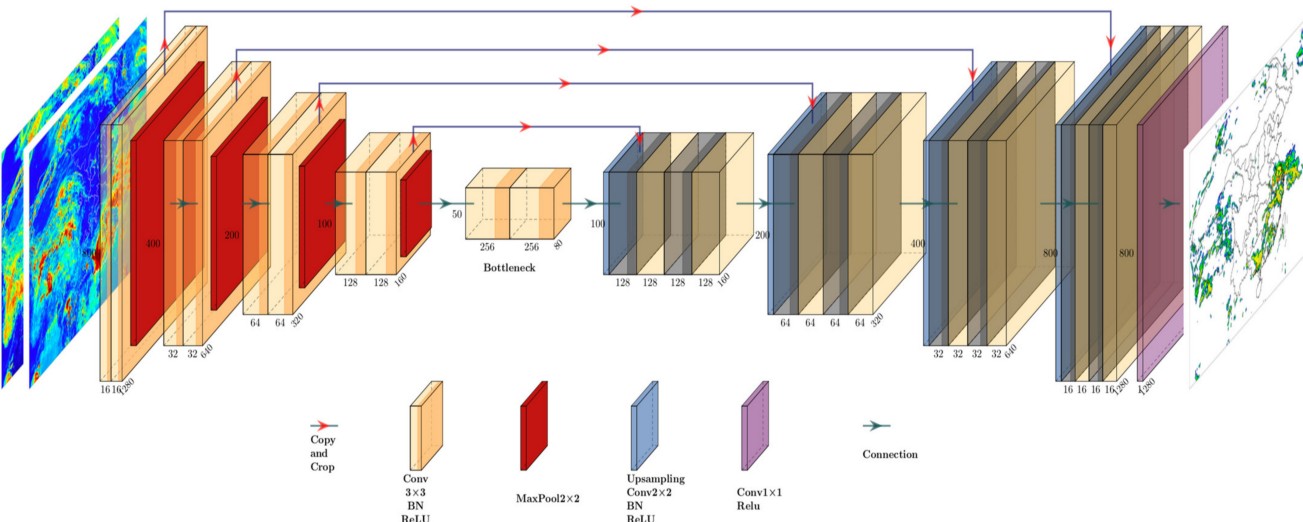

**Figure 2.** A U-net regression architecture (example of 50 × 80 pixels in the lowest resolution). Each colored box corresponds to a multi-channel feature map and the numbers of channels for the feature maps are denoted at the bottom of the box.

### 3.3. Model Training

The network training for the model uses the matched Fengyun-4A/AGRI and RCRF maps based on the Adaptive Moment Estimation of Pytorch (Torch of Python version). To minimize overhead and maximize the use of the Graphics Processing Unit memory, we favor large input tiles over a large batch size to reduce the batch to a single image. Accordingly, we use a high momentum (0.99), such that a large number of the previously seen training samples determine the update in the current optimization step. In addition, the learning rate is set to 0.00016. In this training model, the lost function is computed by a pixel-wise root-mean-square error (RMSE, Equation (2)). Note that only the pixels covered by radar echoes are used to estimate the lost function, as follows.

$$\text{RMSE} = \sqrt{\frac{1}{N}\sum_{i=1}^{N}\left(y_i - y_{pred,i}\right)^2} \tag{2}$$

Also, we introduce the pixel-wise mean absolute error (MAE) and the R-square ($R^2$) (Equations (3) and (4)) as the reference indicators to evaluate the training model.

$$MAE = \sum_{i=1}^{N}\left|y_i - y_{pred,i}\right|/N \tag{3}$$

$$R^2 = 1 - \sum_{i=1}^{N}\left(y_i - y_{pred,i}\right)^2 / \sum_{i=1}^{N}(y_i - \overline{y_i})^2 \tag{4}$$

where $y_i$ and $y_{pred,i}$ are the pixels of the true data and the retrieved RCRF maps, respectively; $\overline{y_i}$ is the average value of the true data. In order to prevent overfitting of the model, we will stop the training step when the verified loss function no longer drops or even rises.

## 4. Validation and Discussions

### 4.1. Case Studies

Figure 3 shows the developing severe convective events covering a large area in China (70°E–135°E, 16°N–55°N) at 08:00 UTC on June 5, 2020. The light blue areas are the regions covered by radar echoes. Obviously, the blind areas of radar echoes significantly affect the real-time monitoring capability on fast-developing severe weather systems that usually propagate from the source to downstream. Particularly, some severe weather systems frequently occur in the area without the coverage of radar echoes, such as part of the Tibetan Plateau, Xinjiang, Inner Mongolia, Northeast China, and coastal areas. As shown in Figure 3, a typical severe convective weather system occurred in the south region of the Yangtze River and South China with heavy precipitation. In the overlapped region, the retrieved radar reflectivity factors match well with the real radar maps. However, in the region outside the coverage of radar echoes, the retrieved radar reflectivity factors also agree well with the precipitation distribution from GPM IMERG data. Although there are no matched samples from blind areas of radar echoes (such as the eastern Japan, the East China Sea, the Tibetan Plateau and areas north of 50°N) for training, the deep learning model can still retrieve a high-quality retrieved radar reflectivity factor as well as areas covered by radar echoes. Thus, these results further demonstrate that the deep learning method can retrieve radar reflectivity factors even in the regions without matched radar samples. The radar reflectivity factors retrieved by infrared and visible models are almost consistent. However, there are differences in details, especially in the severe convective precipitation area (centered at 30°N, 130°E). The details of radar reflectivity factors from convective precipitation predicted by the visible model are more obvious than those by the infrared model.

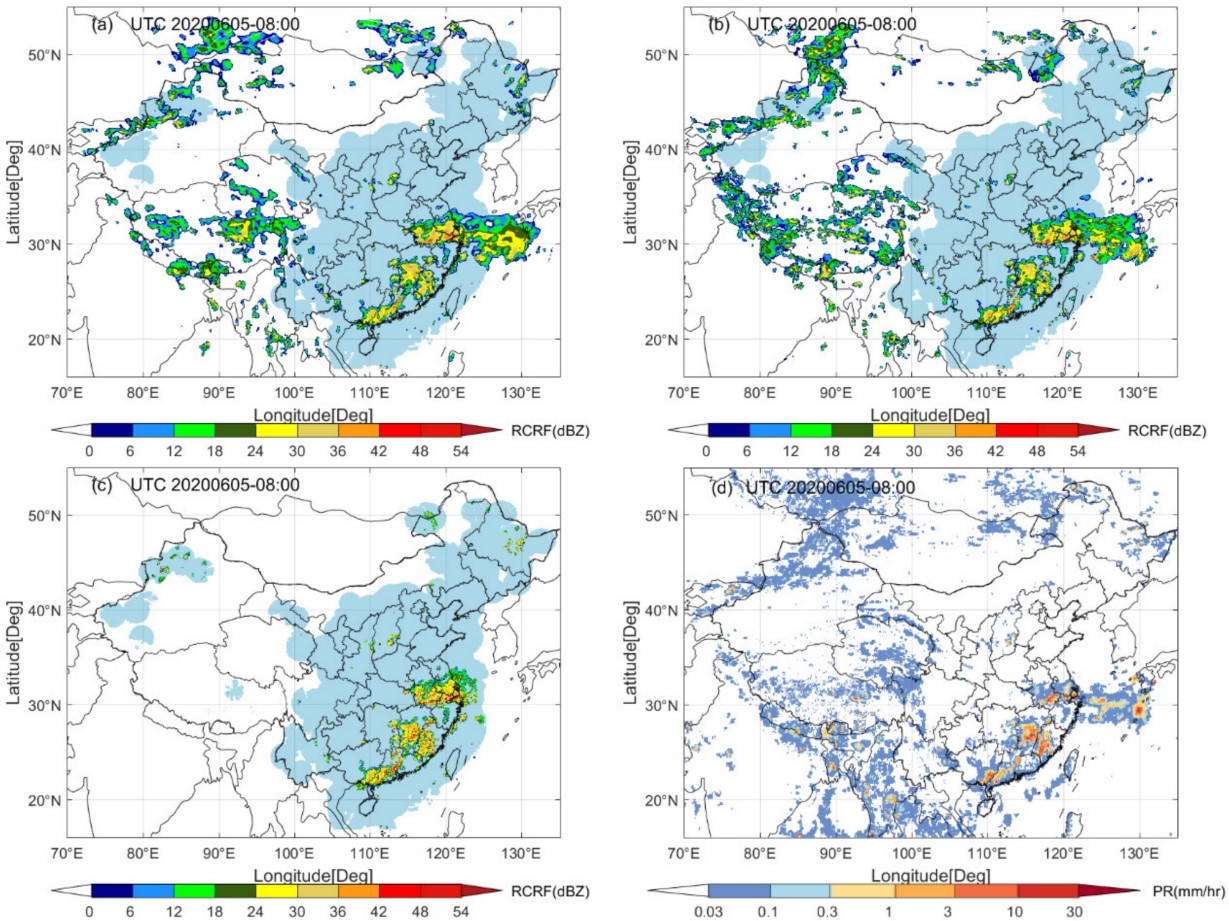

**Figure 3.** (**a**) RCRF maps (dBZ) retrieved from infrared channels, (**b**) visible channels and (**c**) ground-based radars, and (**d**) the corresponding precipitation rate from the GPM IMERG data (mm·h$^{-1}$) at 08:00 UTC on June 5, 2020. The light blue areas are the regions covered by radar echoes.

In addition to convective systems, tropical cyclones are also one of the most destructive disasters in nature [52]. Tropical cyclones originate and develop in ocean areas, which cannot be detected by ground-based radars. Figure 4 shows the case of Super Typhoon Haishen at 02:30 UTC on September 6, 2020. At that time, the spiral cloud systems around the typhoon were affecting Japan and South Korea with heavy precipitation. The eye region of the typhoon can be clearly seen from the RCRF maps retrieved by both infrared and visible models mentioned above. In addition, it can be found that the radar reflectivity on the right side of the typhoon is greater than that on its left side. Note that the RCRF maps retrieved by the infrared model show a better agreement with the precipitation distribution than that retrieved by the visible model. However, the RCRF maps retrieved by the visible model show more details on the cloud systems of Super Typhoon Haishen. Such a situation is primarily attributed to the finer texture of cloud systems that can be clearly seen by the visible band with a higher spatial resolution. For capturing the characteristics of the dynamic and intensity of the typhoon over the ocean area, it would be helpful to diagnose and forecast the intensity and track of the typhoon by using the RCRF maps with a high spatial (of 4 km) and temporal resolution (of 5–15 min).

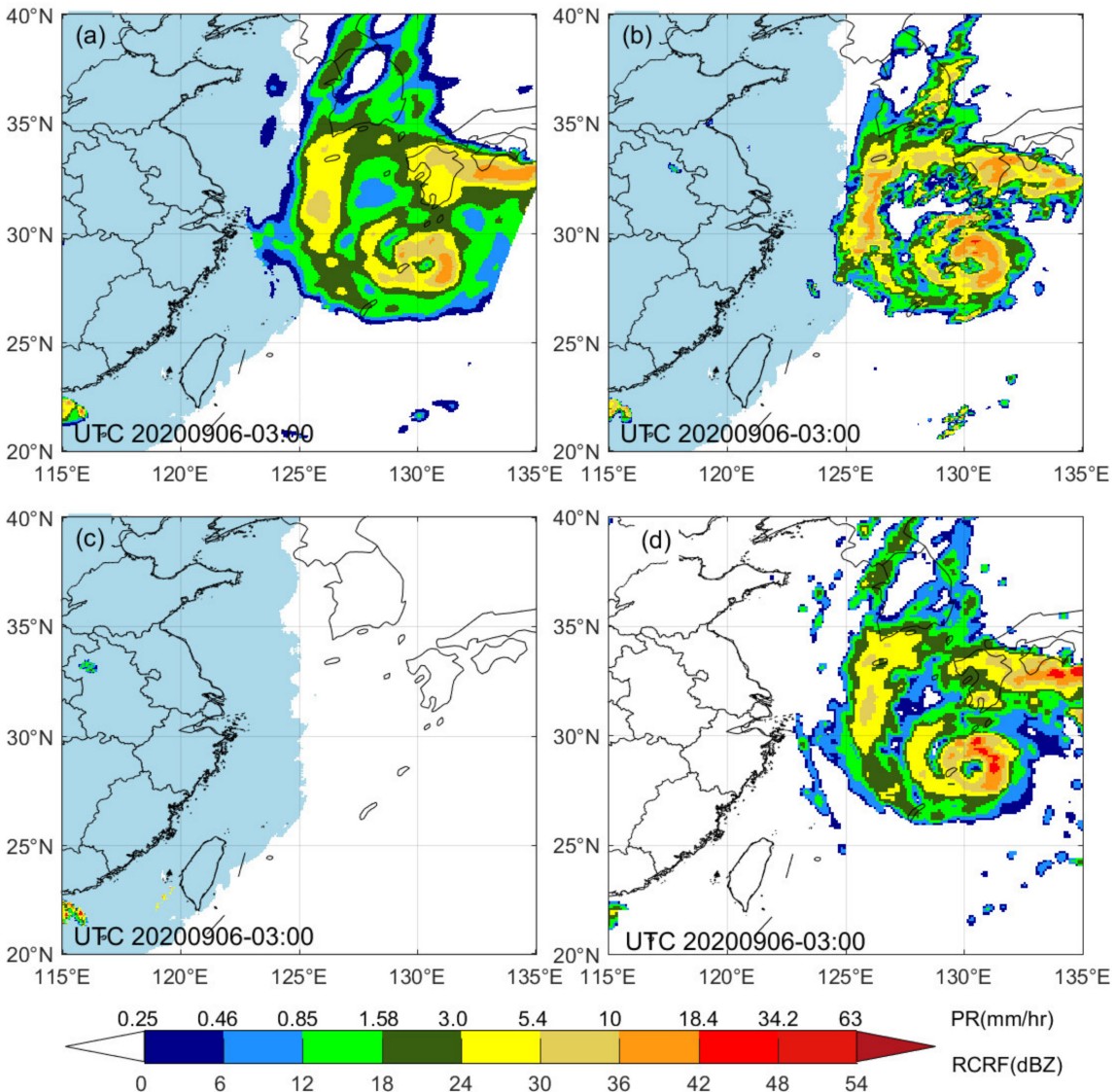

**Figure 4.** A typical case of Super Typhoon Haishen at 3:00 UTC on September 6,2020. (**a**) RCRF maps (dBZ) retrieved from infrared channels, (**b**) visible channels and (**c**) ground-based radars, and (**d**) the corresponding precipitation rate from the GPM IMERG data (mm·h$^{-1}$). The light blue areas are the regions covered by radar echoes.

The Northeast China cold vortex is a common disastrous weather system in the middle and high latitudes of East Asia [53]. Attributed to the disasters induced by rainstorms, thunderstorms, and tornadoes in the mesoscale vortex rain belts, the Northeast China often suffers from great damage and casualties. However, neither Inner Mongolia nor the northeastern provinces of China are fully covered by radar observations. Note that, due to the sparse distribution of radar data in Northeast China, there are some problems in the radar data mosaics in this area. However, this only accounts for a small part of the training samples, and thus it will not affect the overall training effect. Figure 5 shows a typical case of a Northeast China cold vortex that occurred on September 16, 2020. Under the influence of the cold vortex, moderate rain, local heavy rain, and even rainstorm occurred in the east and northeast parts of Inner Mongolia. Figure 5 clearly show the rain belt and intensity distribution from the RCRF maps retrieved by the visible model. In contrast, the value range of the retrieved RCRF maps retrieved by the infrared model is narrower. The RCRF maps from infrared model are closer to the actual observation, while the results from the visible model are higher than the observation. Although the radar mosaic is sparse in

this area, the retrieved RCRF maps are consistent with the precipitation distribution. This conclusion also indicates the robustness of deep learning algorithm.

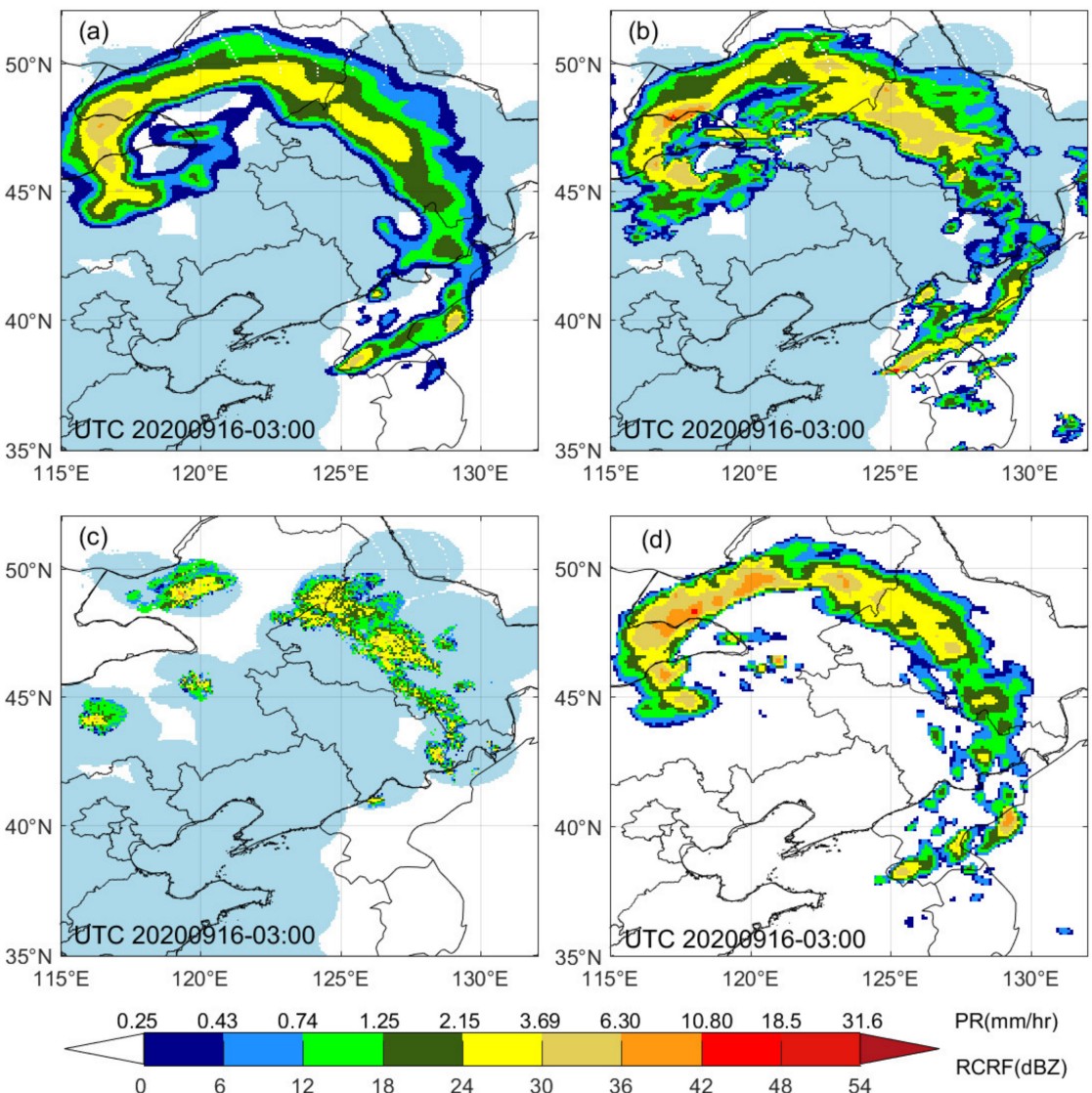

**Figure 5.** A typical case of a Northeast China cold vortex occurred on September 16, 2020. (**a**) RCRF maps (dBZ) retrieved from infrared channels, (**b**) visible channels and (**c**) ground-based radars, and (**d**) the corresponding precipitation rate from the GPM IMERG data (mm·h$^{-1}$). The light blue areas are the regions covered by radar echoes.

Figure 6 presents a set of developing convective events monitored by the retrieved RCRF maps over a time series. This rapidly developing convective system was generated at 04:30 UTC on July 8, 2020, and was maintained for two and a half hours. Then, this convective system entered the mature and stable stage. The retrieved RCRF (especially by the visible model) and precipitation maps at 04:30 UTC show that the convective target-1 (marked by the red circles) has developed to a mature stage, but it only corresponds to 2–3 pixels on radar maps at that moment. Note that similar phenomena are observed for convective target-2, target-3, and target-4. Therefore, it can be concluded that the retrieved RCRF maps in Figure 6 agree with the structures of the precipitation fields very well during the lifespan of this convective system. Particularly, the RCRF maps retrieved by the visible model are more sensitive to the initial convection with long leading time.

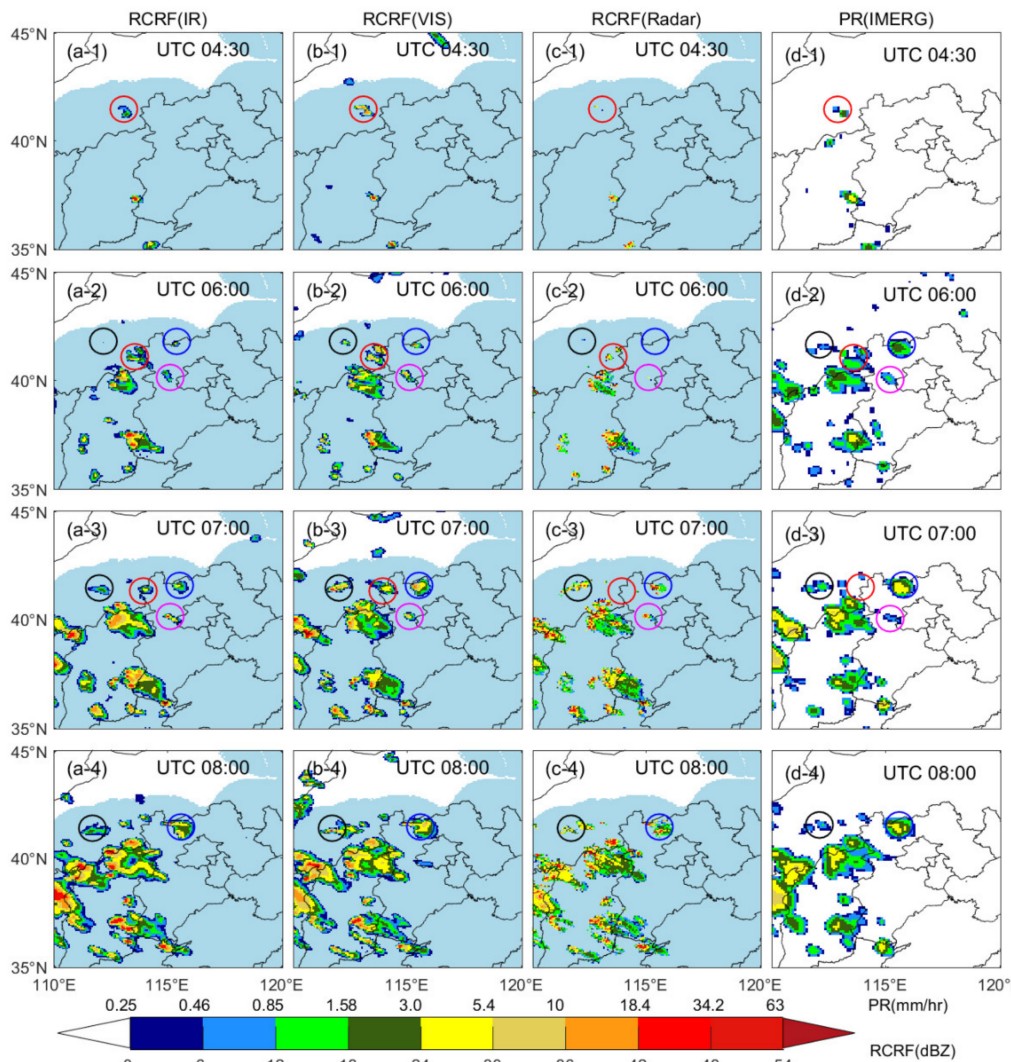

**Figure 6.** The occurrence and development for the case of a convective system occurring over North China from 04:30 UTC to 08:00 UTC on July 8, 2020. The subplots at each row of the panel are RCRF maps (dBZ) retrieved from (**a**) infrared channels, (**b**) visible channels, and (**c**) ground-based radar, and (**d**) the corresponding precipitation rate from the GPM IMERG data (mm·h$^{-1}$). The red circles stand for target-1, the pink circles for target-2, the blue circles for target-3 and the black circles for target-4. The light blue areas are the regions covered by radar echoes.

### 4.2. Statistical Results

In this section, we present the validation results of the RCRF maps retrieved by the visible and infrared models over the land and ocean, respectively. As mentioned in Section 3.1, 15% of labeled datasets are utilized for model validation. As shown in Figure 7, the RCRF maps retrieved by both visible and infrared models over the land and ocean are validated against the ground-based radar data. It is not surprising that more samples concentrate around the 0 dBZ. Generally, the retrieved RCRF maps show a better consistency with the actual situation over the ocean and land, particularly for the cases by visible models.

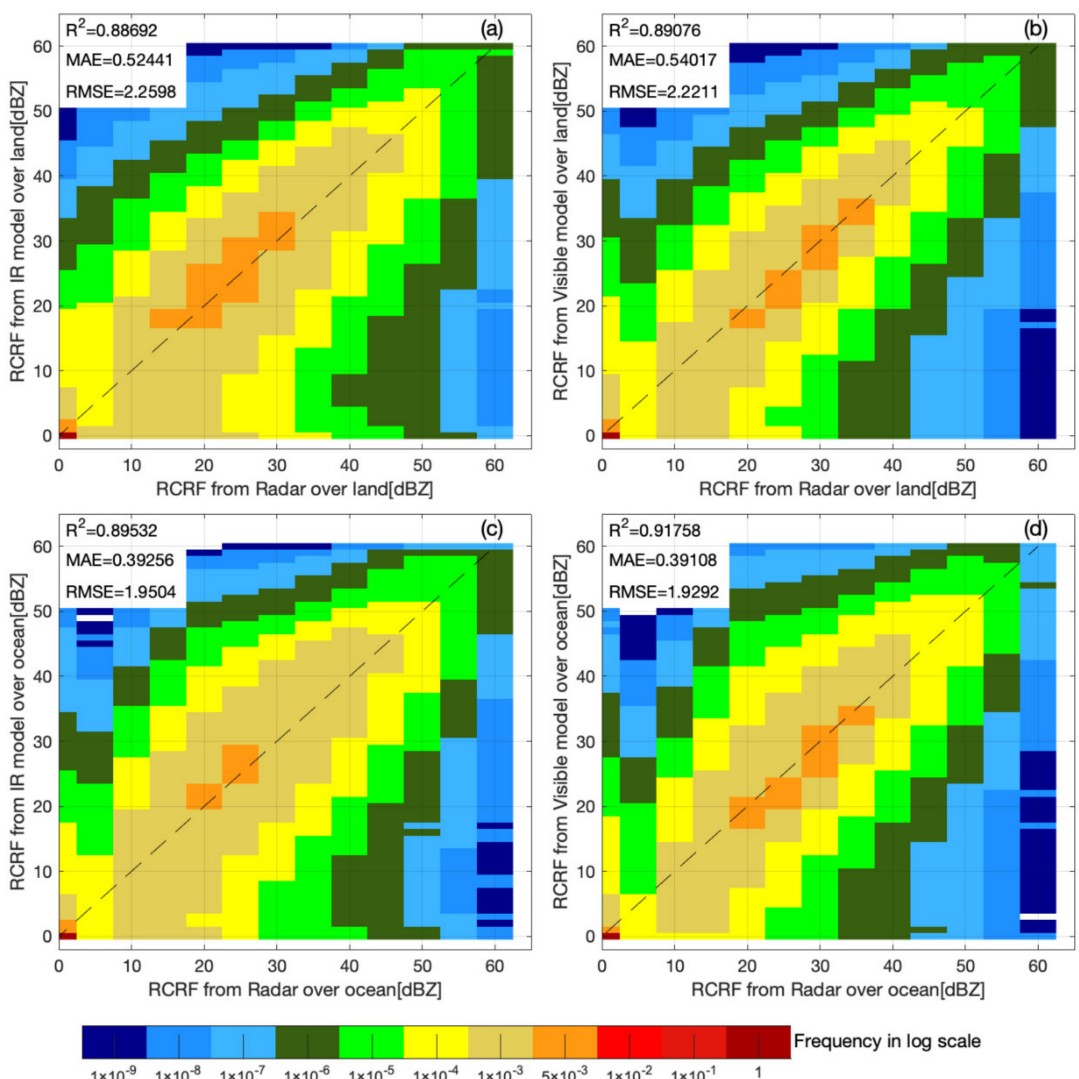

**Figure 7.** Comparisons of the RCRF maps between ground-based radars and deep learning-based retrieving model. The color bar represents the occurrence frequency (in logarithmics scale) for the retrieved RCRF maps. (**a**) The infrared model over the land, (**b**) the visible model over the land, (**c**) the infrared model over the ocean, (**d**) the visible model over the ocean.

Figure 8 presents the statistical indicators for the retrieved RCRF maps over two different regions (land and ocean) from May to October in 2020. It can be seen that the RCRF maps retrieved by the visible model exhibit a better performance over the ocean area, with an MAE of 0.3–0.62 dBZ and an RMSE of 1.6–2.3 dBZ. In contrast, the results from the infrared model over the ocean have an MAE of 0.38–0.7 dBZ and a higher RMSE of 1.8–2.5 dBZ. However, the results from the infrared model over the land performs the worst, with an MAE of 0.4–0.8 dBZ and an RMSE of 1.7–2.8 dBZ. Among them, the worst case occurred in July with an MAE of 0.8 dBZ and an RMSE of 2.8 dBZ. Overall, the accuracy of the retrieved data is higher over the ocean than that over the land in May–July and September. However, the situation in August is the opposite. In addition, the accuracies of the retrieved data over the ocean and land are almost equal in October. One of the reasons is that summer is the peak season for the occurrence and development of severe convections in China. Due to the rapid development of convective systems, the uncertainty of the spatio-temporal matching increases, leading to the relative errors of summer samples. The mechanisms, duration, and spatio-temporal scales of summer precipitation are extremely complicated. In addition, the topography is more complex over the land, where convective systems in summer are mainly triggered by the unstable environment due to the obvious

heating, while the situation is the opposite over the ocean. The complex topography of the land surface together with the more homogeneous ocean surface contribute to the existing differences between land and ocean.

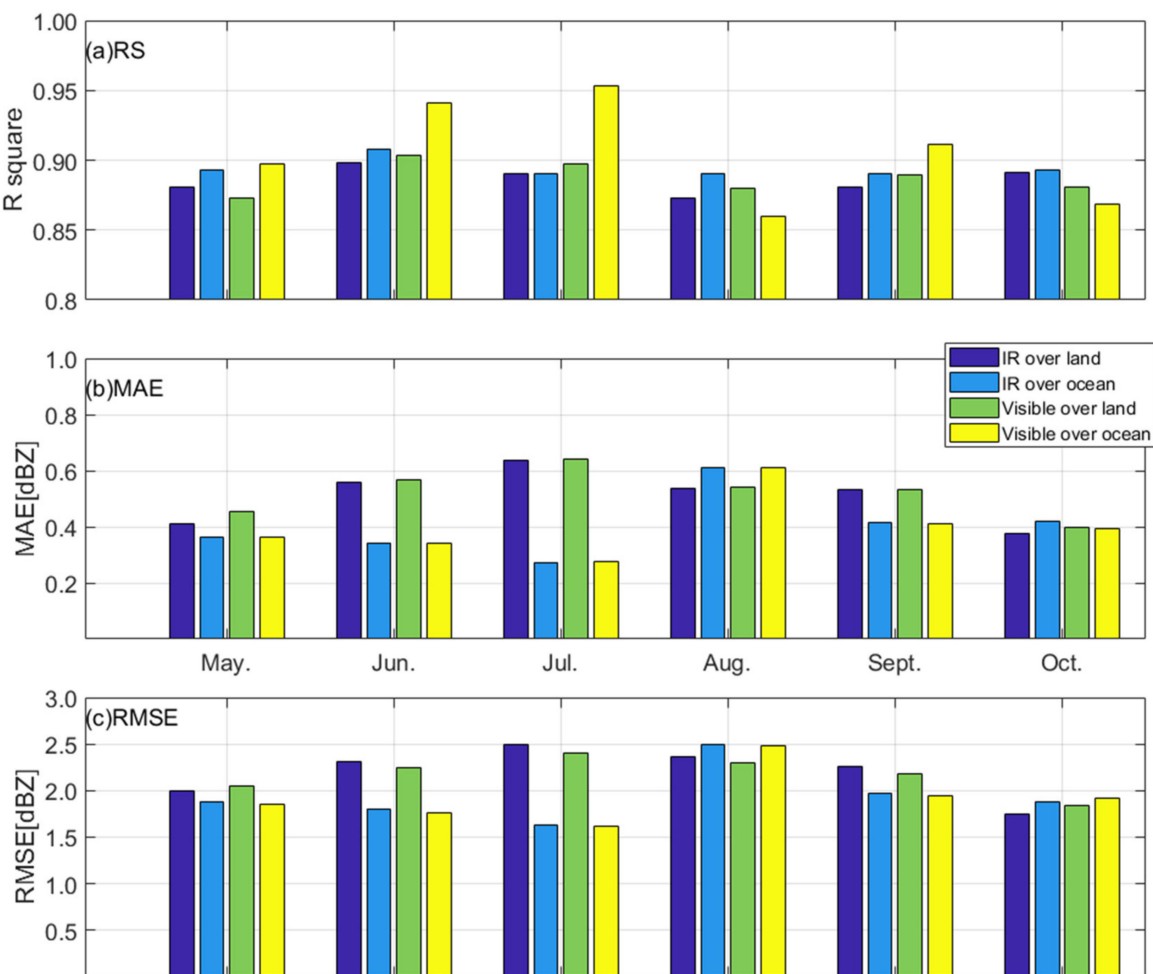

**Figure 8.** Validations on the RCRF obtained by the U-net regression-based retrieving algorithm, for land and ocean regions during May–October, respectively.(**a**) pixel-wise RS, (**b**) pixel-wise MAE, (**c**) pixel-wise RMSE.

### 4.3. Validation Based on Precipitaiton Observations

To validate the retrieved RCRF results using rain/snow measurements, this study also refers to some research on precipitation issues/surface observations [54,55]. The precipitation rate maps have been obtained from the retrieved RCRF maps by using the standard Z–R relationship (see Equation (5)).

$$Z = aR^b \tag{5}$$

where a = 700, b = 1.38. The GPM IMGER datasets are the validating source.

Figure 9 shows a heavy precipitation process in Hubei, Jiangxi, Anhui and JiangSu provinces at 02:00 UTC on July 27,2020 and the specific precipitation rate maps from the retrieved RCRF maps by using standard Z–R relationship.

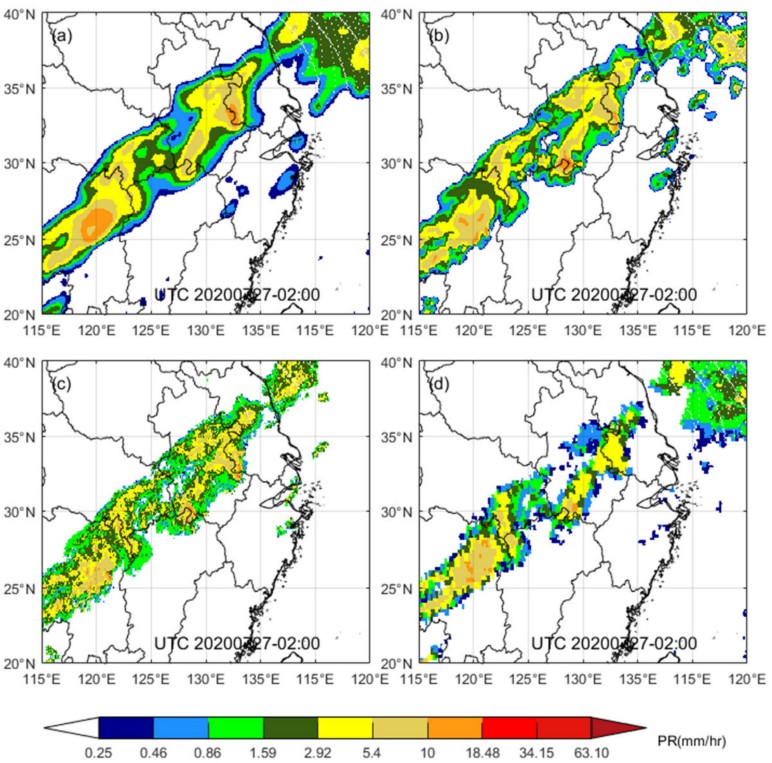

**Figure 9.** Precipitation Rain rate (mm/h) maps at 02:00 UTC on July 27 of 2020, including (**a**) PR retrieved from RCRF maps using infrared channels, (**b**) PR retrieved from RCRF maps using visible/near-infrared channels, (**c**) PR from radar RCRF maps, and (**d**) PR from GPM IMGER data.

The original validating datasets are translated into precipitation rate by using standard Z–R relationships. The matched GPM IMGER datasets are also selected as the validating source of precipitation rate maps. Table 3 presents the statistical indicators for the precipitation rate data retrieved from retrieved RCRF maps by using the standard Z–R relationship from May to October in 2020. From Table 3, it can be seen that the precipitation rate maps retrieved by using the visible model exhibit a better performance with an MAE of 0.247–0.339 mm/h and an RMSE of 0.506–0.701 mm/h, respectively. In contrast, the results from the infrared model have an MAE of 0.259–0.452 mm/h and an RMSE of 0.543–0.991 mm/h, respectively. However, the precipitation rate maps retrieved by both the visible model and infrared model exhibit the worst performance on August.

**Table 3.** Validations on precipitation rate from retrieved RCRF maps by using standard Z-R relationship.

| Application Month | Model I | | Model II | |
|---|---|---|---|---|
| | MAE (mm/h) | RMSE (mm/h) | MAE (mm/h) | RMSE (mm/h) |
| May. | 0.259 | 0.543 | 0.251 | 0.552 |
| Jun. | 0.329 | 0.759 | 0.247 | 0.506 |
| Jul. | 0.385 | 0.776 | 0.274 | 0.524 |
| Aug. | 0.452 | 0.991 | 0.339 | 0.701 |
| Sept. | 0.394 | 0.813 | 0.265 | 0.517 |
| Oct. | 0.279 | 0.594 | 0.297 | 0.637 |

## 5. Conclusions

For severe weather monitoring and nowcasting, this study aims to investigate and develop a unified retrieving algorithm for quantitatively estimating the RCRF maps by China's new generation GEO meteorological satellite Fengyun-4A/AGRI. A novel deep

learning, U-net regression model, is used to estimate near-real-time RCRF maps. This new algorithm is remarkably different from statistical regression methods, and its key advantage is the powerful ability to capture non-linear associational patterns between predictors and predictees, with multi-scale structural characteristics. This study also proposes a comparison test between visible and infrared models for sensitivity analysis. The results show that visible models exhibit a better performance over the ocean area with an MAE of 0.3–0.62 dBZ and an RMSE of 1.6–2.3 dBZ. In contrast, the results from the infrared model over the ocean have an MAE of 0.38–0.7 dBZ and a higher RMSE of 1.8–2.5 dBZ. The worst case is results from the infrared model over the land with an MAE of 0.4–0.8 and an RMSE of 1.7–2.8 dBZ. In addition to the advantages of spatial resolution, another reason for the visible model over the infrared model may be that the cloud optical thickness in the visible channel intuitively reflects the development of cloud and convections.

Moreover, three typical cases including a cluster of rapidly developing convective systems, a Northeast China cold vortex and the Typhoon Haishen, are analyzed and studied. The comparisons reveal that the proxy radar reflectivity obtained by the conversion of Fengyun-4A/AGRI data can capture the strong precipitation signal in areas with insufficient coverage of radar echoes. The RCRF maps retrieved by the visible model are more sensitive to the initial convections with long leading time.

As of this writing, the RCRF map-retrieving algorithm has been under the test of the Feng Yun GEO Algorithm Test-bed [56], and will be integrated into the operational Fengyun-4 operational products system. Ongoing improvements to the RCRF retrieving algorithm will include the combination of both weather radar reflectivity data and precipitation data. In the future, Fengyun-4B will add rapid-scan operation with a 1-min scanning mode and a 1-km spatial resolution. It is thus believed that many potential errors will be greatly reduced by using the higher spatio-temporal resolution data from the newer GEO satellite instruments.

**Author Contributions:** Conceptualization, F.S. and D.Q.; methodology, F.S. and D.Q.; software, F.S.; validation, F.S. and B.L.; formal analysis, F.S., M.M. and D.Q.; investigation, F.S., M.M. and D.Q.; resources, F.S. and D.Q.; data curation, F.S.; writing—original draft preparation, F.S.; writing—review and editing, D.Q. and M.M.; visualization, F.S.; supervision, D.Q.; project administration, F.S. and D.Q.; funding acquisition, F.S. and D.Q. All authors have read and agreed to the published version of the manuscript.

**Funding:** This research was funded by the National Key R&D Program of China under Grants 2017YFB0502803, the Natural Science Foundation of China under Grants 41975031, and the Guangdong Province Key Laboratory for Climate Change and Natural Disaster Studies (Grant 2020B1212060025).

**Institutional Review Board Statement:** Not applicable.

**Informed Consent Statement:** Not applicable.

**Data Availability Statement:** The data presented in this study are available on request from the corresponding author.

**Acknowledgments:** We appreciate the ground-based radar reflectivity data generously shared by Chinese National Meteorological Information Center. Last but not least, we would also like to thank the anonymous reviewers for their thoughtful and constructive suggestions and comments.

**Conflicts of Interest:** The authors declare no conflict of interest.

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
