# Peer review of "Deep Learning-Based Radar Composite Reflectivity Factor Estimations from Fengyun-4A Geostationary Satellite Observations"

_remotesensing, doi:10.3390/rs13112229_

Round 1

Reviewer 1 Report

General comments

This manuscript discusses retrieving the ground radar composite reflectivity factor (RCRF) from Fengyun-4A geostationary satellite observations. The concept of this research paper is interesting, and this kind of model development is needed in this field, especially hurricane track predictions. This paper clearly describes the data collection, preprocessing, and methodology.  I recommend accepting this manuscript with major revision.

Major comments

  1. In section 3. What is the meaning of “ two modules of model training”. Please provide more details.
  2. In Figure 1, the right-side flow chart, what is the inputs data box mean? The validation data set is input to the validation part and it is coming from the left side flow chart  Inputs data box makes more confusion. Please give more details.
  3. Please add few images from the training data set. Please show side-by-side images from the satellite radiances from all channels and corresponding radar RCRF.
  4. Please give basic details (number of radars used, radar frequency etc..) about the radars used in this study.
  5. The size of the training data set looks very small to learn this kind of complex relationship. If possible, please provide a detailed explanation that justifies this data set size is enough to learn the complex relationship between satellite channel radiances and ground radar RCRF. 
  6. Table 2 mentions model I and model II, but it not clearly mentioned in the manuscript what is model I and model II mean. 

Minor comments

1)Page 1, Line number 13: saving peoples’ it should be people’s

2)Page 1, Line number 25: Put a space before dBZ

3)Page 2, Line number 47: First time used numerical weather prediction Please give an abbreviation as NWP and use after that. 

4)Page 2, Line number 52: the cloud , further. It should be the cloud, further

5)Page 2, Line number 71. deep neural network, use abbreviation DNN and use after that,

6)Page 2. Line number 74, such as Graphics Processing Unit, uses abbreviation (GPU).

Reviewer 2 Report

please improve the paper based on precip observations, not only by radar reflectivity.

At least show how the results can be validated using rain/snow measurements etc. I assume no snow precip but discuss it in a separate  discussion section.

I suggest 2 papers for precip issues/surface observations e.g., Gultepe et al (AMS Met Monog 2018) and Pure and applied geopy for aviation supersite observations 2019. 

Overall, nicely written, but needs some discussions, and separating conclusions from discussions.

Reviewer 3 Report

The manuscript is very interesting and solves an important problem with widespread applications especially in nowcasting severe weather. The structure of the manuscript is well organized, however, in my opinion the English should be improved.

I understand that the basic idea was to prepare “radar” data in areas where they are not available and apply to them a nowcasting technique. Please say it clearly if it is true, because when you use “forecast” it is not clear what you have in mind.

Specific comments:

Line 95: I have never seen “remainder” in this context.

Section 3.2: Could you add the number of free parameters whose values are found by “learning”.  

Figure 3 and others: You compare results in dBz with precipitation from IMERG data. It is clear that structures of the fields agree very well but what is the result of quantitative comparison? Could you say something about precipitation obtained by some standard Z-R technique?

Line 303: What do you mean by “dynamic range”?

Line 318: The sentence “Therefore …” is not clear to me. What is clear response?

Line 437: It should be: Sokol, Z.

Line 489-490: I think that a source (journal) is missing.

Round 2

Reviewer 1 Report

I thank the authors for responding to all my comments. I am fine with the response and recommending to accept the paper in the present form.